# High level of soluble tumor necrosis factor receptors is associated with lower residual diuresis volume in patients on hemodialysis: An exploratory study

Gabriele Teixeira Gonçalves[1,2,3], Luciana Martins de Mello Santos[1,2,3,4], Pedro Henrique Scheidt Figueiredo[1,2,3,4], Jaqueline de Paula Chaves Freitas[1,3,5,6], Jousielle Márcia Santos[1,2,3], Joyce Noelly Vitor Santos[1,2,3,5,6], Fidelis Antônio da Silva Junior[1,2,3], Frederico Lopes Alves[3,7,8], Vanessa Gomes Brandão[3,7,8], Emílio Henrique Barroso Maciel[3,7,8], Maria Cecília S. M. Prates[3,4,7,8], Borja Sañudo[9,10], Redha Taiar[11], Mario Bernardo-Filho [12], Danúbia da Cunha de Sá-Caputo[12], Vanessa Pereira Lima[1,2,3,4], Henrique Silveira[1,3,4,5], Victor Lacerda Gripp[1,3], Vanessa Amaral Mendonça[1,3,4,5,6], Ana Cristina Rodrigues Lacerda[1,2,3,4,6]*

1 Centro Integrado de Pós-Graduação e Pesquisa em Saúde (CIPq-Saúde), Universidade Federal dos Vales do Jequitinhonha e Mucuri, Diamantina, Brazil, 2 Laboratório de Fisiologia do Exercício – LAFIEX – CIPq/Saúde, Universidade Federal dos Vales do Jequitinhonha e Mucuri, Diamantina, Brazil, 3 Universidade Federal dos Vales do Jequitinhonha e Mucuri, Diamantina, Brazil, 4 Programa de Pós-Graduação em Reabilitação e Desempenho Funcional, Universidade Federal dos Vales do Jequitinhonha e Mucuri, Diamantina, Brazil, 5 Laboratório de Inflamação e Metabolismo – LIM – CIPq/Saúde, Universidade Federal dos Vales do Jequitinhonha e Mucuri, Diamantina, Brazil, 6 Programa de Pós-Graduação Multicêntrico em Ciências Fisiológicas, Universidade Federal dos Vales do Jequitinhonha e Mucuri, Diamantina, Brazil, 7 Escola de Medicina, Universidade Federal dos Vales do Jequitinhonha e Mucuri, Diamantina, Brazil, 8 Unidade de Hemodiálise, Hospital Santa Casa de Caridade de Diamantina, Diamantina, Brazil, 9 Department of Physical Education and Sports, Universidad de Sevilla, Seville, Spain, 10 Université de Reims Champagne Ardenne, Reims, France, 11 Laboratório de Vibrações Mecânicas, Policlínica Universitária Piquet Carneiro, Instituto de Biologia Roberto Alcantara Gomes, Universidade do Estado do Rio de Janeiro, Rio de Janeiro, Brazil, 12 Programa de Pós-Graduação em Ciências da Saúde, Universidade Federal dos Vales do Jequitinhonha e Mucuri, Diamantina, Brazil

* lacerda.acr@ufvjm.edu.br

## Abstract

### Objective

Patients on hemodialysis commonly present with elevated inflammatory markers. It is noteworthy, however, that higher levels of these markers may deteriorate residual renal function in in these individuals. Further investigation is essential to clarify the potential link between systemic chronic inflammatory parameters and residual diuresis volume in this population, particularly when accounting for confounding variables such as body composition. This study aimed to explore the possible relationship between inflammatory parameters and residual diuresis volume in patients on hemodialysis.

**Data availability statement:** All relevant data are within the article and its Supporting Information files.

**Funding:** The author(s) received no specific funding for this work.

**Competing interests:** The authors have declared that no competing interests exist.

## Methods

Blood samples were collected from patients on hemodialysis for the analysis of soluble receptors: 1) tumor necrosis factor receptor 1 (sTNFR1), 2) tumor necrosis factor receptor 2 (sTNFR2), and 3) leptin. Confounding variables, such as gender, age, duration of hemodialysis, Kt/V (a measure of dialysis adequacy), and body composition assessed using the dual-energy X-ray absorptiometry (DXA), were also evaluated. Data analyses were conducted using both single and multiple regression models, adjusted for the confounding parameters.

## Results

Of the total sixty participants, 27 (45%) were classified as anuric, and 33 (55%) as non-anuric. High sTNFR1 plasma levels were associated with a lower residual diuresis volume, irrespective of adjustments for confounding parameters ($R^2$ = 25.4%; β = 0.504; $p < 0.001$).

## Conclusion

This study supports the hypothesis that higher systemic levels of sTNFR1 may deteriorate residual renal function, as evidenced by the lower residual diuresis volume observed in patients on hemodialysis. These findings suggest that interventions aimed at reducing systemic inflammation may be beneficial in preserving residual renal function and improving clinical outcomes in these patients.

Chronic Kidney Disease (CKD) is characterized by any persistent alteration or abnormality in kidney structure or function lasting over ninety days [1]. CKD's subtle onset complicates diagnosis, and once established, it often progresses irreversibly [2]. Recognized globally as a major public health issue [3], CKD affects 10–13% of adults in economically advanced countries [1]. In Brazil, CKD poses significant healthcare challenges, causing over 35,000 deaths annually and incurring substantial treatment costs [4]. Projections indicate around 10 million CKD cases in Brazil, with about 90,000 individuals requiring dialysis [5,6]. This data underscores the urgent need to address CKD as a critical health challenge and develop comprehensive strategies to mitigate its impact on public health and healthcare costs.

## Background

According to the 2022 Brazilian Dialysis Census, hemodialysis (HD) is the predominant renal replacement therapy in Brazil, used by 95.3% of patients undergoing treatment [7,8]. In 2018, the number of HD patients was 133,464, marking a three-fold increase since 2000 [9].

Residual renal function is crucial for maintaining fluid and electrolyte balance and eliminating toxins in patients undergoing hemodialysis (HD). However, the

deterioration of renal function is a significant concern, as it is associated with poorer clinical outcomes [10]. Our previous work found the absence of residual diuresis in patients on hemodialysis is associated with a higher risk of sarcopenia and low functional performance [11]. In addition, physical activity and Parathormone levels are determinant protagonists of functioning in HD [12].

The uremic condition in patients with CKD often leads to protein and energy depletion, resulting in decreased lean body mass and increased fat mass [13]. This can exacerbate systemic chronic inflammation, a key determinant in the progression of renal failure [1,10,14]. CKD promotes tissue damage through redox imbalance, fibrosis, and cellular apoptosis, leading to the release of cytokines such as tumor necrosis factor (TNF), which are processed by renal function [10,15]. Both tumor necrosis factor receptor 1 (sTNFR1) and tumor necrosis factor receptor 2 (sTNFR2), located on cell surfaces, mediate the activity of TNF, a key pro-inflammatory cytokine involved in inflammatory and immune responses in various diseases [16,17]. Elevated levels of TNF and its soluble receptors correlate with increased mortality in patients on HD [18,19].

Evidence indicates that leptin receptors are distributed across various body organs and systems, influencing physiological functions like bone mass quality, systemic chronic inflammation, and stress response. Leptin, a hormone secreted by adipose tissue, interacts with the hypothalamus to regulate body fat, energy expenditure, satiety, and appetite [20]. Notably, leptin may exacerbate low-grade systemic inflammation by inducing the release of cytokines such as TNF, which are processed by renal function [15].

This study was conducted to investigate the hypothesis that elevated systemic chronic inflammation may deteriorate residual renal function, as evidenced by lower residual diuresis volume in patients on hemodialysis. Although previous studies have evaluated the relationship between systemic inflammatory markers and residual diuresis volume in hemodialysis patients, no studies have been found that investigated this relationship considering body composition measured with precise techniques and taking into account confounding factors such as gender, age, hemodialysis duration, Kt/V, and body composition. Therefore, this study aimed to explore the potential association between systemic inflammatory markers and residual diuresis volume in hemodialysis patients, considering these confounding factors.

## Materials and methods

The study employed an exploratory cross-sectional design and was conducted from October to December 2019 in HD sector of Santa Casa de Caridade Hospital in Diamantina, Minas Gerais, Brazil. Ethical approval was granted by the Human Research Ethics Committee at Universidade Federal dos Vales do Jequitinhonha e Mucuri under protocol number 3,612,157, in accordance with the principles of the Declaration of Helsinki. Written informed consent was obtained from all participating volunteers.

Were included individuals aged 18 years or older diagnosed with CKD who were undergoing HD treatment at least three times per week for more than six months. Patients who were taking non-steroidal anti-inflammatory drugs or corticosteroids were excluded.

Patients underwent HD sessions using low-flux dialysis membranes, which are crucial as they can influence the efficacy of the treatment and the patients' immune response. Water treatment involved reverse osmosis and ultraviolet light disinfection processes, emphasizing the importance of water quality in preventing infectious complications and ensuring patient safety during dialysis. Unfractionated heparin was used for anticoagulation during the HD sessions, with adjustments based on activated partial thromboplastin time (aPTT) to carefully manage anticoagulation and avoid thrombotic complications.

The sample size was calculated using GPower Software (version 3.1.9.2). A pilot study was conducted with 15 HD patients to evaluate various predictors, including age, duration of HD treatment, Kt/V ratio, body weight, body composition metrics, bone mineral density, and inflammatory markers. With an effect size of 0.78 for residual diuresis volume, a 5% alpha error, and a statistical power of 98%, the required sample size was determined to be 60 patients.

### Procedures

**Experimental design.** The study incorporated two sessions conducted on dialysis days, collecting data before and after HD sessions.

**Initial session.** The patient assessment included anamnesis focusing on personal and disease-related factors such as gender, age, comorbidities, HD duration, and residual diuresis volume [12].

**Residual diuresis volume measurement.** Participants measured their residual diuresis using a graduated container, collecting urine 24 hours preceding the first weekly HD session [19]. This procedure is standard in the HD sector, with which patients are already familiar. Based on the volume collected, patients were classified as anuric (less than 100 mL/day) or non-anuric (more than 100 mL/day) (20). Dialysis efficacy, indicated by Kt/V indexes, was assessed according to the standards set by the National Kidney Foundation [9].

**Blood biomarkers.** The blood sample was collected by the nursing team from the HD sector at Hospital Santa Casa de Caridade de Diamantina during the puncture of the arteriovenous fistula at the beginning of the second weekly HD session. Two 10 mL tubes of blood samples without anticoagulant were collected from each patient. They were subsequently centrifuged at 3000 rpm for 10 minutes and serum, plasma, and hemolysate samples were separated and stored at −80°C until analysis. To analyze the plasma levels of sTNFR1, sTNFR2, and Leptin, thawed samples were promptly tested using an enzyme-linked immunosorbent assay (ELISA) kit (Quantikine, R&D Systems, Minneapolis, MN) following the manufacturer's instructions. The ELISA method was employed and reactions were read at 490 nm using a microplate reader (SoftMax Pro, version 2.2.1, Softmax, Sunny Valley, CA). The detection limits for these assays were 5 pg/mL for leptin, sTNFR1, and sTNFR2.

**Dual energy x-ray absorptiometry (DEXA).** Utilizing equipment from Lunar Radiation Corporation (Madison, Wisconsin, USA, model DPX), this tool determined body weight, body fat, android/gynoid ratio, visceral fat, fat mass, fat mass index (FMI), lean mass, lean mass index (LMI), height, and bone mineral density at the total hip and lumbar spine (L2-L4). FMI and LMI were calculated by dividing the fat mass and lean mass, respectively, by the square of the height. Assessments were carried out immediately after the second weekly HD session to minimize any impact from excess tissue fluid [20].

**Statistical analysis.** Data were analyzed using SPSS version 22.0 (SPSS Inc., Chicago, IL, USA). The Shapiro-Wilk and Levene tests assessed the sample distribution and homoscedasticity, respectively. Descriptive statistics characterized the sample [19]. Continuous data were presented as means with a 95% confidence interval, while categorical variables were presented as absolute numbers and percentages [19]. Group comparisons between anuric and non-anuric patients were made using t-tests or Mann-Whitney tests [19]. Simple linear regression was applied to independent variables with $p < 0.20$ in correlation tests (Pearson or Spearman), and stepwise multiple linear regression models were developed for outcomes with $p < 0.10$ in simple regressions [19]. Adjustments were made for age, fractional urea clearance, and hemodialysis vintage, with a significance level set at 5% [12].

**Multiple regression analysis.** Four assumptions were verified: linearity, even distribution of residuals, homoscedasticity, and absence of multicollinearity [12]. Linearity and residual distribution were checked using scatter plots and histograms, respectively. Homoscedasticity was confirmed through scatter plots showing an even distribution along the regression line, and absence of multicollinearity was verified with VIF values below 10.0 [21]. The Durbin-Watson test checked for autocorrelation, with acceptable values ranging from 1.5 to 2.5 [12].

## Results

### Patient selection and characteristics

Of the 128 patients undergoing HD, 109 met the inclusion and 49 were excluded criteria, with 60 subjects ultimately evaluated (62% men; mean age 54.00 years [95% CI: 50.15–57.85]) (Fig 1). All eligible patients underwent hemodialysis through an arteriovenous fistula. Additionally, the study characterized HD patients' body composition—including lean

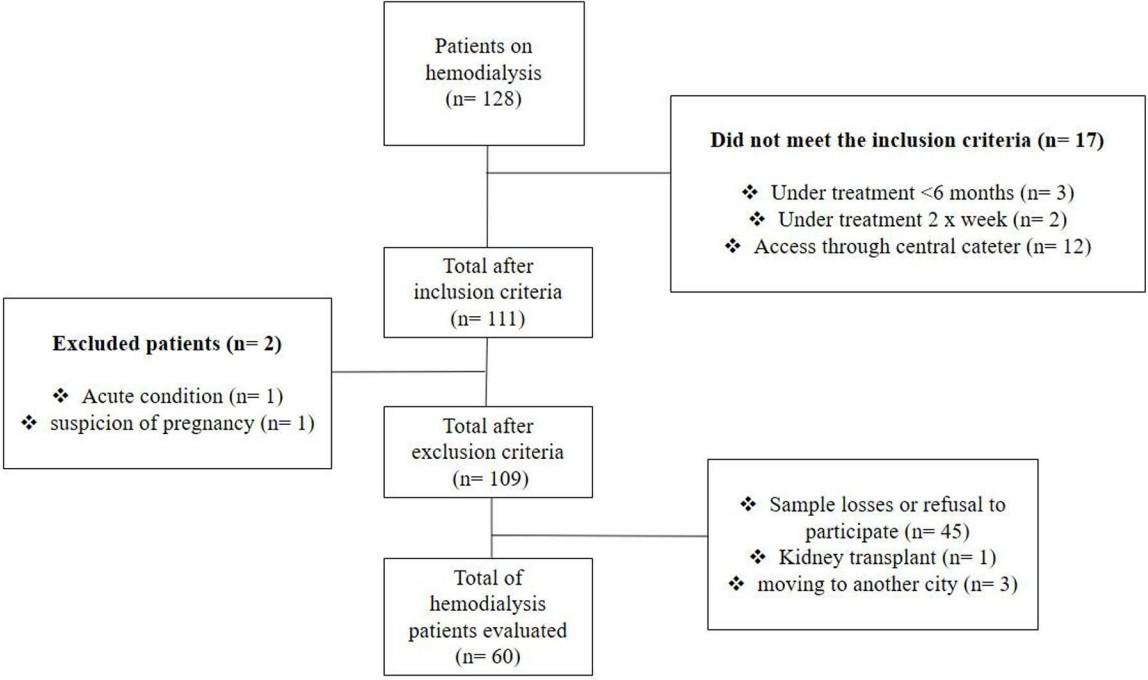

**Fig 1. Flowchart of inclusion and exclusion criteria.**

mass, fat mass, and bone mineral density—and systemic biomarkers (Table 1).The raw dataset containing these data is available in the Supporting Information (S1 Dataset).

## Patient classification and biochemical analysis

Out of the evaluated patients, 27 (45%) were classified as anuric, and 33 (55%) as non-anuric. Although anuric patients underwent HD treatment for a longer duration, no significant differences were observed in other clinical variables, such as body composition and bone mineral density. Plasma leptin concentrations were consistent across groups. However, higher concentrations of soluble receptors, specifically sTNFR1 and sTNFR2, were detected in the anuric group (Table 2).

## Correlation and association analysis

The analysis revealed a moderately negative correlation between daily residual diuresis volume and hemodialysis vintage, as indicated in Table 3. A weak positive association was found between daily residual diuresis volume and the android/gynoid (a/g) ratio, visceral fat, fat mass, and fat mass index, according to DXA measurements. Notably, there was a weak association with sTNFR2 and a moderately negative correlation with sTNFR1. No statistically significant associations were observed with other anthropometric, clinical, or personal characteristics

## Regression analysis

Regression analysis indicated that daily residual diuresis volume had a weak negative association with HD duration, a weak positive association with the a/g ratio, a moderate negative association with sTNFR1, and a weak negative association with sTNFR2. Notably, a high plasma level of sTNFR1 was central catheterd with a lower residual diuresis volume, accounting for 25.4% of the variability in residual diuresis volume, even after adjusting for confounding factors such as gender, age, HD duration, Kt/V, and body composition (Table 4).

**Table 1. Patients sample characteristics.**

| Sample characteristics (n=60) | n (%) | Mean (95% CI) |
|---|---|---|
| Gender *(%)* | | |
| Male | 37 (62) | |
| Female | 23 (38) | |
| Age *(years)* | | 54.00 (50.15–57.85) |
| Diuresis Volume *(mL)* | | 456.58 (269.44–643.73) |
| Duration of HD *(years)* | | 4.70 (3.55–5.85) |
| Kt/v | | 1.42 (1.33–1.52) |
| Body fat *(%)* | | 28.85 (26.20–31.50) |
| a/g ratio | | 0.98 (0.88–1.07) |
| Visceral fat *(g)* | | 919.34 (696.94–1141.74) |
| Fat mass *(kg)* | | 18.48 (15.85–21.11) |
| FMI *(kg/m²)* | | 7.15 (6.02–8.28) |
| Lean mass *(kg)* | | 43.07 (40.38–45.75) |
| LMI *(kg/m²)* | | 16.36 (15.76–16.96) |
| BMD total *(g/cm³)* | | 0.99 (0.95–1.04) |
| BMD spine *(g/cm³)* | | 1.00 (0.93–1.08) |
| BMD hip *(g/cm³)* | | 0.83 (0,78–0.87) |
| Leptin *(ng/mL)* | | 870.60 (705.96–1035.24) |
| sTNF-R1 *(pg/mL)* | | 2821.07 (2581.15–3061.00) |
| sTNF-R2 *(pg/mL)* | | 2129.35 (2037.74–2220.96) |

Duration of HD: duration of hemodialysis; Kt/V: fractional urea clearance; a/g ratio: android/gynoid ratio; FMI: fat mass index; LMI: lean mass index; BMD: bone mineral density; sTNF-R1: 1 soluble receptor of tumor necrosis factor alpha; sTNF-R2: 2 soluble receptor of tumor necrosis factor alpha.

**Table 2. Comparison between residual urine outputs.**

| Variables | Anurics (n=27) | Diuresis volume ≥ 100 mL (n=33) | p-value |
|---|---|---|---|
| Age *(years)* | 50.37 ± 14.97 | 56.97 ± 14.40 | 0.088 |
| Duration of HD *(years)* | **6.17 (5.04–8.82)** | **2.00 (1.78–4.10)** | **0.000*** |
| Kt/v | 1.48 ± 0.25 | 1.37 ± 0.44 | 0.263 |
| Body fat *(%)* | 26.82 ± 9.74 | 30.51 ± 10.52 | 0.168 |
| a/g ratio | 0,72 (0.62–1.13) | 1.02 (0.73–1.28) | 0.097 |
| Visceral fat *(g)* | 481.00 (148.00–1277.00) | 784.50 (336.50–1745.00) | 0.136 |
| Fat mass *(Kg)* | 14.68 (11.65–17.92) | 19.92 (16.10–23.27) | 0.076 |
| FMI *(kg/m²)* | 5.23 (4.56–7.42) | 6.74 (6.31–9.58) | 0.139 |
| Lean mass *(kg)* | 42.46 ± 9.85 | 43.48 ± 9.98 | 0.713 |
| LMI *(kg/m²)* | 15.98 ± 2.00 | 16.62 ± 2.31 | 0.291 |
| BMD total *(g/cm³)* | 0.96 ± 0.19 | 1.02 ± 0.15 | 0.184 |
| BMD spine *(g/cm³)* | 0.96 (0.83–1.16) | 1.02 (0.91–1.19) | 0.232 |
| BMD hip *(g/cm³)* | 0.81 ± 0.21 | 0.84 ± 0.17 | 0.626 |
| Leptin *(ng/mL)* | 431.51 (166.26–1330.36) | 1195.85 (384.44–1567.92) | 0.076 |
| sTNF-R1 *(pg/mL)* | **3263.49 ±877.02** | **2435.74 ± 765.36** | **0.000*** |
| sTNF-R2 *(pg/mL)* | **2239.60 ± 315.42** | **2033.32 ± 352.03** | **0.023*** |

Data present by mean ± standard deviation or median (quartile 1–quartile 3). Duration of HD: duration of hemodialysis; Kt/V: fractional urea clearance; a/g ratio: android/gynoid ratio; FMI: fat mass index; LMI: lean mass index; BMD: bone mineral density; sTNF-R1: 1 soluble receptor of tumor necrosis factor alpha; sTNF-R2: 2 soluble receptor of tumor necrosis factor alpha.

**Table 3. Correlations according to the sample distribution of patients on hemodialysis.**

| Chronic Kidney Disease Patients on Hemodialysis (mean/DP) | Diuresis Volume, mL | |
|---|---|---|
| | r | p |
| Age (years) | 0.184 | 0.160 |
| Duration of HD (years) | −0.490 | 0.000* |
| Kt/v | −0.196 | 0.136 |
| Body fat (%) | 0.172 | 0.189 |
| a/g ratio | 0.307 | 0.017* |
| Visceral fat (g) | 0.305 | 0.026* |
| Fat mass (Kg) | 0.286 | 0.042* |
| FMI (Kg/m²) | 0.271 | 0.047* |
| Lean mass (Kg) | 0.042 | 0.761 |
| LMI (Kg/m²) | 0.099 | 0.474 |
| BMD total (g/cm³) | 0.074 | 0.572 |
| BMD spine (g/cm³) | 0.101 | 0.442 |
| BMD hip (g/cm³) | 0.083 | 0.528 |
| Leptin (ng/mL) | 0.258 | 0.053 |
| sTNF-R1 (pg/mL) | −0.484 | 0.000* |
| sTNF-R2 (pg/mL) | −0.326 | 0.012* |

Duration of HD: duration of hemodialysis; Kt/V: fractional urea clearance; a/g ratio: android/gynoid ratio; FMI: fat mass index; LMI: lean mass index; BMD: bone mineral density; sTNF-R1: 1 soluble receptor of tumor necrosis factor alpha; sTNF-R2: 2 soluble receptor of tumor necrosis factor alpha.

**Table 4. Independent contributors to diuresis volume in patients on hemodialysis (n = 60).**

| Independent Variables | Univariate | | | Multivariate | | | p Value |
|---|---|---|---|---|---|---|---|
| | R² Adjusted | Beta | p Value | R² Adjusted | Beta | | |
| sTNF-R1 (pg/mL) | 0.234 | −0.484 | 0.000* | 0.254 | −0.504 | | < 0.001 |
| sTNF-R2 (pg/mL) | 0.106 | −0.326 | 0.012* | | | | NS |
| Duration of HD (years) | 0.112 | −0.335 | 0.010* | | | | NS |
| a/g ratio | 0.081 | 0.285 | 0.027* | | | | NS |
| Kt/V | 0.039 | −0.196 | 0.136 | | | | NS |
| Body fat (%) | 0.030 | 0.172 | 0.189 | | | | NS |
| Visceral fat (g) | 0.043 | 0.208 | 0.135 | | | | NS |
| Fat mass (Kg) | 0.006 | 0.080 | 0.578 | | | | NS |
| FMI (Kg/m²) | 0.011 | 0.107 | 0.441 | | | | NS |
| Leptin (ng/mL) | 0.019 | 0.139 | 0.301 | | | | NS |
| Age (years) | 0.034 | 0.184 | 0.160 | | | | NS |

sTNF-R1: 1 soluble receptor of tumor necrosis factor alpha; sTNF-R2: 2 soluble receptor of tumor necrosis factor alpha; Duration of HD: duration of hemodialysis; a/g ratio: android/gynoid ratio; Kt/V: fractional urea clearance; FMI: fat mass index.

## Discussion

The primary outcome of this study, which included patients on HD, indicated that elevated levels of soluble tumor necrosis factor receptors (sTNFR), especially sTNFR1, accounted for 25.4% of the variance in lower residual diuresis volume. Remarkably, this association persisted even after adjusting for confounding variables such as gender, age, HD duration,

Kt/V, and body composition. These findings suggest that sTNFR1, a biomarker of tubular damage, may significantly influence residual diuresis volume in patients on HD.

Based on our results and given the innovative approach of assessing body composition using more precise equipment, we have identified a research gap concerning the potential association between inflammatory markers and residual diuresis volume in patients on HD. Individuals with higher levels of circulating plasma markers, such as leptin, sTNFR1, and sTNFR2, might exhibit increased systemic chronic inflammation and reduced residual diuresis. Notably, despite the prevalent systemic inflammatory process associated with CKD and HD, there is still a lack of understanding regarding inflammatory biomarkers that influence the reduction of residual diuresis volume.

Consistent with our findings, previous studies have demonstrated that elevated levels of soluble sTNFR, particularly sTNFR1, are associated with the deterioration of renal function, tubular damage, and an increased risk of all-cause mortality in patients on HD, regardless of age [10,18]. Additionally, higher levels of pro-inflammatory cytokines have been observed in patients with diminished residual renal function and advanced renal disease [22]. In contrast, patients on HD with preserved residual diuresis volume tend to have a better quality of life, reduced levels of inflammation, and higher survival rates [18,23]. Notably, sTNFR1 levels were higher in patients on HD without residual renal function [18]. This aligns with our study's findings, which identified elevated levels of sTNFR1 in the blood of anuric patients. Moreover, previous research indicated that maintaining residual diuresis volume in this population may sustain lower serum concentrations of inflammatory cytokines compared to anuric patients, potentially indicating a less severe inflammatory state [10,24]. Furthermore, preventing or delaying the total loss of GFR can improve survival in these individuals [25].

Localized on cell membranes, sTNFR1 is activated by binding to TNF-alpha, which is produced by various cells of the immune system, initiating inflammatory and stress response pathways [1]. Oxidative stress, driven by the production of reactive oxygen species, plays a crucial role in the development of comorbidities [21]. Experimental studies have shown that TNF-alpha contributes to the pathophysiology of CKD by inducing renal vasoconstriction, which leads to increased superoxide production and a simultaneous decrease in nitric oxide (NO) availability.

In essence, soluble sTNFRs are emerging as promising biomarkers for kidney damage. However, further research is essential to determine the clinical relevance of sTNFR measurements for identifying renal impairments and predicting reduced residual diuresis volume. Preliminary findings from a study involving two independent cohorts highlight the importance of circulating sTNFRs as relevant biomarkers for kidney damage and dysfunction in the elderly, even in the absence of diabetes [26].

While this study did not find significant associations between leptin and residual diuresis volume, previous research has shown that leptin acts as a uremic toxin that contributes to the progression of CKD and its deleterious effects [15,27]. Produced by adipose tissue, leptin is affected by variables such as gender, hormones, and medications. Its elimination, facilitated by renal function, decreases as GFR declines, resulting in elevated serum levels in the presence of inflammation-induced hyperinsulinemia. Hyperleptinemia in patients with kidney failure is associated with pathological changes in the kidneys, including glomerular and tubular damage, leading to significant proteinuria and renal parenchymal fibrosis. Additionally, leptin shows a negative correlation with GFR and a positive correlation with Body Mass Index [27,28].

However, the variability in leptin levels and its impact on CKD have been associated with both a lower risk of all-cause mortality and a higher risk of cardiovascular mortality [27]. Consequently, the precise role of leptin in CKD remains unclear, underscoring the need for further research.

Moreover, body composition—including lean mass, bone mass, and fat mass—can influence or be influenced by systemic inflammation in patients on HD. Including a representative measure of body composition in the analysis is crucial to reduce the risk of biased interpretations. The absence of significant correlations in our analysis was contrary to our expectations, given the assumed link between the inflammatory process observed in this population and aspects of body composition and residual renal function [26].

Previous studies have suggested that patients with residual renal function exhibit better nutritional status, a higher percentage of body fat, and a higher lean mass index. The uremic state that develops in conjunction with the decline in glomerular filtration rate stimulates catabolism, increases protein breakdown, and decreases protein anabolism [29,30].

Patients on HD suffer from poorer muscle function, anabolic resistance, and worse protein and caloric balance compared to patients undergoing other treatments for CKD [31]. Despite the non-significant results, the clinical implications of our findings may be useful. Beyond anuria, other factors such as overall health status, the adequacy of the dialysis procedure, and the management of comorbidities may play important roles.

Additionally, we must consider limitations that may have affected the results, such as sample characteristics. Although the mean age is within the age range of the Brazilian population on dialysis, the sample was younger than that studied by other authors. Therefore, further research is essential to explore other factors influencing body composition in the CKD spectrum, especially in older people. These studies should focus on larger sample sizes and the exploration of other factors that may influence body composition in these individuals longitudinally,

Once residual renal function is recognized as a predictor of survival for patients with CKD, being associated with a substantial reduction in mortality risk, greater vitality, fewer dietary restrictions, and less exacerbation of the inflammatory state and oxidative stress and their consequences, monitoring the residual diuresis volume through a simple and reliable measure, as in our study, can be beneficial for this population. This approach can contribute to reducing costs and preventing additional disease-related damage [10,26,32].

In conclusion, our study supports the hypothesis that lower residual diuresis volume in patients on HD may be associated with elevated systemic levels of sTNFR1. However, this association should not be interpreted as indicative of a causal relationship, and the direction of the association is also uncertain. High levels of inflammatory markers can be a consequence (at least partly) of low residual kidney function rather than the etiology of low residual kidney function, or both. Additional research is necessary to clarify the potential mechanisms that link sTNFR1 to residual diuresis volume in HD patients and examine the implications for patient management and treatment strategies.

## Strengths and limitations

Our investigation stands out for its meticulous characterization of study participants. It offers detailed insights into kidney phenotypes through comprehensive analyses of the duration of HD, Kt/V values, and other relevant confounding factors. We employed gold standard methods for measuring body composition variables such as body fat, lean mass, and bone mineral density rather than relying on less accurate indirect estimates. Despite these strengths, it is essential to acknowledge certain limitations. The generalizability of our findings to different age and ethnic groups is not assured, which may restrict the external applicability of the study results. Additionally, the cross-sectional observational nature of our study limits our ability to establish causality. Furthermore, although we have detailed data on the glomerular filtration rate, the absence of glomerular ultrafiltration rate data may limit a complete understanding of the renal filtration process during hemodialysis sessions and their clinical implications. Nonetheless, our research contributes significantly to the sparse literature in Latin America on the correlation between inflammatory markers and residual diuresis volume. It enriches the existing body of knowledge and lays crucial groundwork for future longitudinal studies that are needed to clarify the cause-and-effect relationships involving the biomarkers studied.

## Conclusion

In conclusion, this study supported the benefits of monitoring residual diuresis volume in hemodialysis patients. Our findings contribute to a limited but growing body of research suggesting that inflammatory biomarkers, such as sTNFR1, can play a role in the progression of end-stage renal disease by triggering inflammatory cascades, which can be associated with residual diuresis volume. Additionally, there is a continuing need for longitudinal studies aimed at developing effective methods for monitoring and preserving residual diuresis volume.

## Supporting information

**S1 Dataset. Raw dataset containing data from 60 participants used for analysis in this study.** The dataset includes the following variables: Height, Lean Mass (kg), LMI (kg/m²), Fat Mass (kg), FMI (kg/m²), Duration of HD (years), Diuresis Volume (mL), Kt/V, Body Fat (%), A/G Ratio, Visceral Fat (g), BMD Total (g/cm³), BMD Spine (g/cm³), BMD Hip (g/cm³), Leptin (ng/mL), sTNF-R1 (pg/mL), and sTNF-R2 (pg/mL).
(PDF)

## Acknowledgments

We would like to express our gratitude to the Universidade Federal dos Vales do Jequitinhonha e Mucuri, National Council for Scientific and Technological Development, Foundation for Research Support of the State of Minas Gerais, and Coordination for the Improvement of Higher Education Personnel for their support and for providing scholarships. Additionally, our sincere thanks go to the Hemodialysis Unit of Hospital Santa Casa de Caridade de Diamantina, along with its collaborators and volunteers who contributed to this research.

## Author contributions

**Conceptualization:** Maria Cecília S M Prates.

**Data curation:** Gabriele Teixeira Gonçalves, Luciana Martins de Mello Santos, Jaqueline de Paula Chaves Freitas.

**Formal analysis:** Gabriele Teixeira Gonçalves, Luciana Martins de Mello Santos, Jaqueline de Paula Chaves Freitas.

**Methodology:** Gabriele Teixeira Gonçalves, Luciana Martins de Mello Santos, Jaqueline de Paula Chaves Freitas.

**Supervision:** Maria Cecília S M Prates.

**Writing – original draft:** Gabriele Teixeira Gonçalves, Luciana Martins de Mello Santos, Pedro Henrique Scheidt Figueiredo, Jaqueline de Paula Chaves Freitas, Jousielle Márcia Santos, Joyce Noelly Vitor Santos, Fidelis Antonio da Silva Junior, Frederico Lopes Alves, Vanessa Gomes Brandão, Emílio Henrique Barroso Maciel, Maria Cecília S M Prates, Borja Sañudo, Redha Taiar, Mario Bernardo-Filho, Danúbia da Cunha de Sá-Caputo, Vanessa Pereira Lima, Henrique Silveira, Victor Lacerda Gripp, Vanessa Amaral Mendonça, Ana Cristina Rodrigues Lacerda.

**Writing – review & editing:** Gabriele Teixeira Gonçalves, Luciana Martins de Mello Santos, Pedro Henrique Scheidt Figueiredo, Jaqueline de Paula Chaves Freitas, Jousielle Márcia Santos, Joyce Noelly Vitor Santos, Fidelis Antonio da Silva Junior, Frederico Lopes Alves, Vanessa Gomes Brandão, Emílio Henrique Barroso Maciel, Maria Cecília S M Prates, Borja Sañudo, Redha Taiar, Mario Bernardo-Filho, Danúbia da Cunha de Sá-Caputo, Vanessa Pereira Lima, Henrique Silveira, Victor Lacerda Gripp, Vanessa Amaral Mendonça.

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
