## [Decision Letter · Decision Letter 0]

11 Oct 2024

PONE-D-24-26287High level of soluble tumor necrosis factor receptors is associated with lower residual diuresis volume in hemodialysis patients: An exploratory study.PLOS ONE

Dear Dr. Lacerda,

Thank you for submitting your manuscript to PLOS ONE. After careful consideration, we feel that it has merit but does not fully meet PLOS ONE’s publication criteria as it currently stands. Therefore, we invite you to submit a revised version of the manuscript that addresses the points raised during the review process.

We look forward to receiving your revised manuscript.

Kind regards,

Te-Ling Lu

Academic Editor

PLOS ONE

Journal Requirements:

2. Please remove all personal information, ensure that the data shared are in accordance with participant consent, and re-upload a fully anonymized data set. 

Reviewers' comments:

Reviewer's Responses to Questions

**Comments to the Author**

1. Is the manuscript technically sound, and do the data support the conclusions?

Reviewer #1: Yes

Reviewer #2: Yes

2. Has the statistical analysis been performed appropriately and rigorously? 

Reviewer #1: No

Reviewer #2: Yes

3. Have the authors made all data underlying the findings in their manuscript fully available?

Reviewer #1: No

Reviewer #2: Yes

4. Is the manuscript presented in an intelligible fashion and written in standard English?

Reviewer #1: Yes

Reviewer #2: Yes

5. Review Comments to the Author

Reviewer #1: While I think an interesting analysis the lack of longitudinal data is regrettable. I think the conclusions drawn from the work are not sufficiently corroborated by the data since it only reports a cross-sectional association between inflammatory markers and residual renal function. Also - the correlational analyses are conducted in the overall sample of 60 HD patients, a study population which also comprises 27 anuric patients. As such the dependent variable is zero for almost half of the population - which I think will very much affected the results. A subset analysis in those with sufficient RRF is indicated but will suffer from the small ample size.

Reviewer #2: This study looked at the relationship between serum inflammation-related molecules and the level of residual kidney function in a cross-sectional cohort of patients on chronic HD.

Comments:

Indicate why only patients who have been on HD for more than 6 months were included.

Indicate why you chose to look at residual kidney function as a binary variable and why the cut-off of 100ml/day. Consider running the analyses with residual kidney function as a continuous variable.

What was the average interval (in days, weeks, months, or years) between the DEXA study and the testing of residual kidney function?

What was the average interval (in days, weeks, months, or years) between the measurement of sTNF-R1 and sTNF-R2 and the testing of residual kidney function?

Indicate the type of vascular access used for HD in Tables 1 and 2. Depending on vascular access type distribution between the groups, include the vascular access in regression analysis and Table 4.

The authors state: “During the study, patients underwent HD sessions using low-flux dialysis membranes,” Clarify what ‘during the study’ means, how long the patients were in the study.

Before entering the study, were any patients on high-flux dialyzers? If so, how many HD sessions, on average, did they receive with high-flux dialyzers before being switched to low-flux dialyzers?

Were other urine tests obtained, such as urea nitrogen or creatinine, and if so, please report Kidney stdKt/V or kidney urea clearance, kidney creatine clearance, in Tables 1 and 2

Consider adding in the Discussion that, besides the fact that this study does not prove causality, the direction of the association is also uncertain, For example, high levels of inflammatory markers can be a consequence (at least partly) of low residual kidney function rather than the etiology of low residual kidney function, or both.

Limitations appear in two separate sections –on page 15 and 16. Needs to be consolidated into one section, before Conclusions,

Same for Conclusions, it needs to be consolidated, it is mentioned on page 15 ‘Inc conclusion’, and again the paragraph on page 16,

Reference 11 – did not cross over properly, missing authors, journal

Tables 3 and 4: a/g ratio: add in the legend what a/g denotes

6. PLOS authors have the option to publish the peer review history of their article (what does this mean? ). If published, this will include your full peer review and any attached files.

**Do you want your identity to be public for this peer review?** For information about this choice, including consent withdrawal, please see our Privacy Policy .

Reviewer #1: No

Reviewer #2: **Yes: ** Mariana Murea

---

## [Author Response · Author response to Decision Letter 1]

30 Oct 2024

Attached are response documents to reviewers.

---

## [Decision Letter · Decision Letter 1]

5 Jan 2025

PONE-D-24-26287R1High level of soluble tumor necrosis factor receptors is associated with lower residual diuresis volume in patients on hemodialysis: An exploratory study.PLOS ONE

Dear Dr. Lacerda,

Thank you for submitting your manuscript to PLOS ONE. After careful consideration, we feel that it has merit but does not fully meet PLOS ONE’s publication criteria as it currently stands. Therefore, we invite you to submit a revised version of the manuscript that addresses the points raised during the review process.

We look forward to receiving your revised manuscript.

Kind regards,

Te-Ling Lu

Academic Editor

PLOS ONE

Reviewers' comments:

Reviewer's Responses to Questions

**Comments to the Author**

1. If the authors have adequately addressed your comments raised in a previous round of review and you feel that this manuscript is now acceptable for publication, you may indicate that here to bypass the “Comments to the Author” section, enter your conflict of interest statement in the “Confidential to Editor” section, and submit your "Accept" recommendation.

Reviewer #2: All comments have been addressed

Reviewer #3: (No Response)

2. Is the manuscript technically sound, and do the data support the conclusions?

Reviewer #2: Partly

Reviewer #3: Yes

3. Has the statistical analysis been performed appropriately and rigorously? 

Reviewer #2: No

Reviewer #3: Yes

4. Have the authors made all data underlying the findings in their manuscript fully available?

Reviewer #2: No

Reviewer #3: Yes

5. Is the manuscript presented in an intelligible fashion and written in standard English?

Reviewer #2: Yes

Reviewer #3: Yes

6. Review Comments to the Author

Reviewer #2: The authors addressed the comments satisfactorily with one exception which regards the following comment:

Comment 5: Indicate the type of vascular access used for HD in Tables 1 and 2. Depending on vascular access type distribution between the groups, include the vascular access in regression analysis and Table 4.

Author’s Response: Thank you for your consideration. The patients utilized either a central catheter or an arteriovenous fistula. While the type of vascular access is known to be associated with the patient's clinical prognosis (Yeh et al., 2019), it does not directly correlate with residual kidney function.

Reviewer’s Recommendations for R1: The type of vascular access used for hemodialysis can significantly impact serum levels of inflammatory markers. Hence, the recommendation to include the vascular access in the regression analysis remains very important.

Reviewer #3: The authors investigated the relationship between inflammatory markers and residual renal function in a cohort of 60 hemodialysis patients, of which 27 (45%) were anuric. Their findings revealed that elevated sTNFR1 levels were linked to reduced residual urine volumes, even after accounting for confounding factors.

The study is well-presented, and its conclusions are robust. Given the significance of these findings in translational research, this article warrants consideration for publication.

7. PLOS authors have the option to publish the peer review history of their article (what does this mean? ). If published, this will include your full peer review and any attached files.

**Do you want your identity to be public for this peer review?** For information about this choice, including consent withdrawal, please see our Privacy Policy .

Reviewer #2: No

Reviewer #3: No

---

## [Author Response · Author response to Decision Letter 2]

6 Feb 2025

Reviewer: 2

Comments:

The authors addressed the comments satisfactorily with one exception which regards the

following comment:

Comment 5: Indicate the type of vascular access used for HD in Tables 1 and 2.

Depending on vascular access type distribution between the groups, include the vascular

access in regression analysis and Table 4.

Author’s Response: Thank you for your consideration. The patients utilized either a

central catheter or an arteriovenous fistula. While the type of vascular access is known to

be associated with the patient's clinical prognosis (Yeh et al., 2019), it does not directly

correlate with residual kidney function.

Reviewer’s Recommendations: The type of vascular access used for hemodialysis can

significantly impact serum levels of inflammatory markers. Hence, the recommendation

to include the vascular access in the regression analysis remains very important.

Response:

We appreciate the reviewer's comments. The review of the database sample revealed that,

although it was not mentioned as an exclusion criterion in this study, none of the eligible

patients were using a catheter. Additionally, a previous study conducted by our group (1)

established arteriovenous fistula access as the sole inclusion criterion, reflecting a

methodological approach adopted by the research group.

The manuscript also included information according to the reviewer's remark. Page 8,

Lines 176-177: All eligible patients underwent hemodialysis through an arteriovenous

fistula.

Reference

1 Santos, L.M.M., Figueiredo, P.H.S., Silva, A.C.R. et al. Determining factors of

functioning in hemodialysis patients using the international classification of functioning,

disability and health. BMC Nephrol 23, 119 (2022). https://doi.org/10.1186/s12882-022-

02719-5

Reviewer: 3

Comments:

The authors investigated the relationship between inflammatory markers and residual

renal function in a cohort of 60 hemodialysis patients, of which 27 (45%) were anuric.

Their findings revealed that elevated sTNFR1 levels were linked to reduced residual urine

volumes, even after accounting for confounding factors.

The study is well-presented, and its conclusions are robust. Given the significance of these

findings in translational research, this article warrants consideration for publication.

Response: Thank you for considering our study for publication.

---

## [Decision Letter · Decision Letter 2]

12 Feb 2025

High level of soluble tumor necrosis factor receptors is associated with lower residual diuresis volume in patients on hemodialysis: An exploratory study.

PONE-D-24-26287R2

Dear Dr. Lacerda,

We’re pleased to inform you that your manuscript has been judged scientifically suitable for publication and will be formally accepted for publication once it meets all outstanding technical requirements.

Kind regards,

Te-Ling Lu

Academic Editor

PLOS ONE

Additional Editor Comments (optional):

Reviewers' comments:

Reviewer's Responses to Questions

**Comments to the Author**

1. If the authors have adequately addressed your comments raised in a previous round of review and you feel that this manuscript is now acceptable for publication, you may indicate that here to bypass the “Comments to the Author” section, enter your conflict of interest statement in the “Confidential to Editor” section, and submit your "Accept" recommendation.

Reviewer #2: All comments have been addressed

2. Is the manuscript technically sound, and do the data support the conclusions?

Reviewer #2: Yes

3. Has the statistical analysis been performed appropriately and rigorously? 

Reviewer #2: Yes

4. Have the authors made all data underlying the findings in their manuscript fully available?

Reviewer #2: Yes

5. Is the manuscript presented in an intelligible fashion and written in standard English?

Reviewer #2: Yes

6. Review Comments to the Author

Reviewer #2: The authors addressed and clarified all suggestions. In particular, they clarified that all patients included in this analysis had a AVF as their vascular access. I have no further suggestions.

7. PLOS authors have the option to publish the peer review history of their article (what does this mean? ). If published, this will include your full peer review and any attached files.

**Do you want your identity to be public for this peer review?** For information about this choice, including consent withdrawal, please see our Privacy Policy .

Reviewer #2: No

---

## [Editor Report · Acceptance letter]

PONE-D-24-26287R2

PLOS ONE

Dear Dr. Lacerda,

I'm pleased to inform you that your manuscript has been deemed suitable for publication in PLOS ONE. Congratulations! Your manuscript is now being handed over to our production team.

Kind regards,

on behalf of

Dr. Te-Ling Lu

Academic Editor

PLOS ONE